# General External Uncertainty Models of Three-Plane Intersection Point for 3D Absolute Accuracy Assessment of Lidar Point Cloud

**Minsu Kim** [1,*], **Seonkyung Park** [2], **Jeffrey Danielson** [3], **Jeffrey Irwin** [3], **Gregory Stensaas** [3], **Jason Stoker** [4] **and Joshua Nimetz** [5]

1    KBR, Contractor to U.S. Geological Survey (USGS) Earth Resources Observation and Science (EROS), Sioux Falls, SD 57198, USA

2    United Support Services (USS), Contractor to USGS EROS, Sioux Falls, SD 57198, USA; seonkyungpark@contractor.usgs.gov

3    USGS EROS, Sioux Falls, SD 57198, USA; daniels@usgs.gov (J.D.); jrirwin@usgs.gov (J.I.); stensaas@usgs.gov (G.S.)

4    USGS, National Geospatial Program, Reston, VA 20192, USA; jstoker@usgs.gov

5    USGS, National Geospatial Technical Operations Center, Denver, CO 80225, USA; jnimetz@usgs.gov

*    Correspondence: minsukim@contractor.usgs.gov

**Abstract:** The traditional practice to assess accuracy in lidar data involves calculating RMSEz (root mean square error of the vertical component). Accuracy assessment of lidar point clouds in full 3D (three dimension) is not routinely performed. The main challenge in assessing accuracy in full 3D is how to identify a conjugate point of a ground-surveyed checkpoint in the lidar point cloud with the smallest possible uncertainty value. Relatively coarse point-spacing in airborne lidar data makes it challenging to determine a conjugate point accurately. As a result, a substantial unwanted error is added to the inherent positional uncertainty of the lidar data. Unless we keep this additional error small enough, the 3D accuracy assessment result will not properly represent the inherent uncertainty. We call this added error "external uncertainty," which is associated with conjugate point identification. This research developed a general external uncertainty model using three-plane intersections and accounts for several factors (sensor precision, feature dimension, and point density). This method can be used for lidar point cloud data from a wide range of sensor qualities, point densities, and sizes of the features of interest. The external uncertainty model was derived as a semi-analytical function that takes the number of points on a plane as an input. It is a normalized general function that can be scaled by smooth surface precision (SSP) of a lidar system. This general uncertainty model provides a quantitative guideline on the required conditions for the conjugate point based on the geometric features. Applications of the external uncertainty model were demonstrated using various lidar point cloud data from the U.S. Geological Survey (USGS) 3D Elevation Program (3DEP) library to determine the valid conditions for a conjugate point from three-plane modeling.

**Keywords:** lidar; 3D accuracy assessment; external uncertainty model

## 1. Introduction

As point densities of airborne lidar datasets increase, the need for full 3D absolute accuracy assessments of the associated lidar point clouds is becoming more important. Casual users of lidar data may mistakenly believe that higher point density data directly correlate with higher accuracy data. Lidar accuracy is a direct function of the error balance inherent in the system and its operation [1]. Hodgson and Bresnahan [2] described and subset lidar error into four components. In decreasing order

of importance, these included lidar system measurements, interpolation error, horizontal displacement error, and survey error.

To properly assess the absolute accuracy of airborne lidar data, one must have ground truth data that are of higher quality and accuracy than the data being tested. Usually this involves using survey-grade global positioning system (GPS) checkpoints across the project. According to the American Society for Photogrammetry and Remote Sensing (ASPRS) Accuracy Standards for Digital Geospatial Data [3], the independent source of higher accuracy for checkpoints shall be at least three times more accurate than the required accuracy of the geospatial dataset being tested.

The ASPRS Accuracy Standards for Digital Geospatial Data also define methods for assessing vertical and horizontal accuracy for lidar [3]. According to ASPRS, vertical accuracy shall be tested by comparing the elevations of the surface represented by the dataset with elevations determined from an independent source of higher accuracy. This is done by comparing the elevations of the checkpoints (usually collected using survey-grade GPS) with elevations interpolated from the dataset at the same *x/y* coordinates. Vertical accuracy is then computed using root mean square error (RMSE) statistics in non-vegetated terrain and 95th percentile statistics in vegetated terrain.

The non-vegetated vertical accuracy (NVA) at the 95% confidence level in non-vegetated terrain is approximated by multiplying the accuracy value of the vertical accuracy class (or RMSEz) by 1.9600. This calculation includes survey checkpoints located in traditional open terrain (bare soil, sand, rocks, and short grass) and urban terrain (asphalt and concrete surfaces). The NVA, based on the RMSEz multiplier, should be used only in non-vegetated terrain where elevation errors typically follow a normal error distribution. RMSEz-based statistics should not be used to estimate vertical accuracy in vegetated terrain or where elevation errors often do not follow a normal distribution [3].

The vegetated vertical accuracy (VVA) at the 95% confidence level in vegetated terrain is computed as the 95th percentile of the absolute value of vertical errors in all vegetated land cover categories combined, including tall weeds and crops, brush lands, and fully forested areas. For all vertical accuracy classes, the VVA standard is 3.0 times the accuracy value of the vertical accuracy class. Both the RMSEz and 95th percentile methodologies specified above are currently widely accepted in standard practice and have been proven to work well for typical elevation datasets derived from current technologies [3].

While extremely useful for bare earth terrain data derived from lidar, these standards do not fully address the full three-dimensional (3D) accuracy of a lidar point cloud. As a result of the non-transparent and sometimes empirical calibration procedures employed by system manufacturers and data providers, collected lidar data might exhibit systematic discrepancies between conjugate surface elements in overlapping strips that could affect overall data accuracy [4]. By comparing overlapping strips from the same instrument, Habib et al. [4] diagnosed the systematic errors inherent in the instrument. Habib et al. [4] assessed the relative accuracy of the adjusted lidar point cloud but they did not assess the absolute accuracy compared to survey-grade ground truth data. Hebel and Stilla [5] used a combination of a region-growing approach with a random-sample-consensus segmentation method to extract planar shapes in overlapping areas to determine both the boresight parameters and the data alignment. Keyetieu and Seube [6] used simple line patterns over regular slopes in overlapping strips for boresighting.

Another method investigated to calibrate data involves fitting data to planes. The method proposed in Skaloud et al. [7] estimates the calibration parameters by conditioning a group of points to lie on a common plane. The planes were selected manually. The calibration procedure estimated the system parameters as well as the parameters describing the involved planes. The proposed procedure could only be applied whenever planar surfaces with varying slopes and aspects were available. Huang et al. [8] extracted check points from two artificial line segments or three planes for accuracy assessment.

Other research has documented the use of more than simple GPS points to assess the accuracy of airborne data. Tulldahl et al. [9] quantified local surface smoothness on planar surfaces, and distance and relative height accuracy from an unmanned aircraft system (UAS) data using data from a

terrestrial laser scanner as a reference. Cheng et al. [10] proposed an approach for registering airborne and terrestrial laser scanning data at building corners. Canavosio-Zuzelski et al. [11] used a raised hexagonal retro-reflective lidar ground target (HRRT) to address a point location uncertainty and other researches utilized the geometric targets from airborne or mobile lidar [12,13].

For the geometric feature extraction from overlapping swaths, an interactive closest point (ICP) method [14] and its variants [15,16] extract 3D plane based on frame-to-frame correspondence registration. Plane finding techniques from the lidar point cloud are available based on Hough transform [17], region growing [18–20], and the random sample consensus (RANSAC) algorithm [21]. From the accuracy assessment perspective, specific interest is on the roof plane extraction algorithms. Rottensteiner et al. [22] described a plane segmentation method for roof line delineation based on the local homogeneity of surface normal vector. Kim and Shan [23] used the level set principles, Awrangjeb and Fraser [24] used coplanarity and locality of points, and Demir [25] used RANSAC in segmenting roof planes. Dal Poz and Fernandes utilized high-resolution areal image along with lidar data in automatic extraction of building roof boundaries [26].

With these other example methods in mind, this paper investigates geometric feature-based methods to identify a conjugate point from "ground truth data" represented in the airborne lidar point cloud for accuracy assessment. However, the uncertainty associated with the geometric feature-based conjugate point identification can vary. As a result, we investigate what are the preferred geometric features and the valid conditions for identifying the conjugate points. This paper documents extensive airborne lidar simulation modeling with a large array of pyramid targets in order to estimate the uncertainty associated with identifying a conjugate point. We call this attribute external uncertainty. We have developed a generalized external uncertainty model for the three-plane intersection point identification process. We also demonstrate the practical use of the general external uncertainty model using several example lidar point cloud datasets, both real and simulated. The development of the external uncertainty model is a crucial component in establishing a foundation of the 3D absolute accuracy assessment of the lidar point cloud.

## 2. Conjugate Point Identification

The positional uncertainty assessment of lidar point clouds can be accomplished by comparing the 3D ($x$, $y$, $z$) coordinates of many surveyed checkpoints to their conjugate points represented in airborne lidar. The differences between ground-surveyed values and those conjugate point coordinates identified in the airborne lidar point cloud can be summarized into horizontal and vertical statistical uncertainties. Thus, to do this properly, it is important to determine the conjugate point in the airborne lidar point cloud.

### 2.1. Direct Conjugate Point Identification

Consider a ground checkpoint ($x0$, $y0$, $z0$) surveyed on the corner of a paint mark for a pedestrian walkway Figure 1a. A simulated airborne lidar point cloud was created with color displayed by intensity Figure 1b. Since these points are on a flat road surface, if display color is based on elevation, it is not possible to see any visual difference between white paint return points and background road return. Thus, color by intensity is used. In this example, the rainbow color scheme has red representing the high intensity return point and purple for the low intensity return. Since it is a simulated point cloud, the paint boundary and the true conjugate corner point is seen and marked as a blue dot Figure 1b. However, due to the low simulated point density, which is 9 PPSM (points per square meter), it is difficult to identify the true conjugate point location. The conventional practice is to ignore horizontal errors in the direct georeferencing of the airborne lidar data, and simply use horizontal coordinates of the ground checkpoint, represented in this example as a yellow dot ($x0$, $y0$). The interpolated z-value for the ($x0$, $y0$) is compared to the ground truth ($z0$). The vertical differences ($z$–$z0$) are collected from a statistically meaningful number of checkpoints to compute RMSEz.

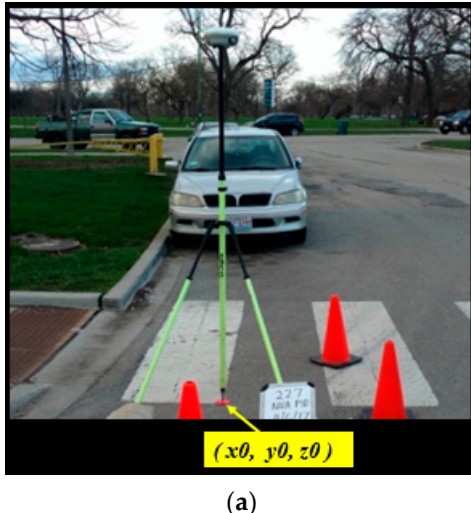
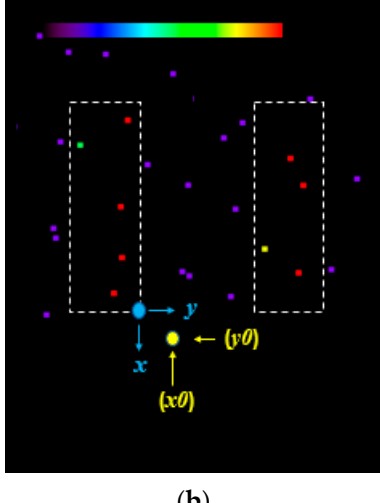

(**a**) (**b**)

**Figure 1.** Ground checkpoint and a simulated airborne lidar point cloud: (**a**) Surveyed coordinates at the corner of the paint mark. (**b**) Yellow dot is the point using checkpoint coordinate, blue dot is the ground truth point in the simulated airborne data using a known simulated paint mark boundary.

These simulated data demonstrate that it is challenging to pinpoint the blue dot from the low point density data. In fact, 9 PPSM is better than the U.S. Geological Survey (USGS) quality level 1 (QL1) requirement [27]. Based on these results, it seems difficult to make any attempt for data at QL2 or lower. A solution would be to increase the point density. However, even simulated 100 PPSM data have 10 cm average point spacing, which adds 5 cm (half of the point spacing) point identification uncertainty in addition to the inherent uncertainty. To overcome this difficulty, a solution is to rely on the point determined from a geometrical feature.

*2.2. Geometric Feature-Based Conjugate Point Identification*

While identifying an isolated conjugate point is a process with high uncertainty, finding a conjugate point based on geometrical features is somewhat easier to achieve. Although geometrical feature-based approaches are useful in general, we need to further investigate the types of geometric features. For example, the points in Figure 2a are the types of conjugate points that are difficult to identify, while the points in Figure 2b are easier to identify. In the example from Figure 1, we illustrated that the external point of any polygon is hard to identify. Similarly, the end point of an isolated linear feature is hard to determine, unless the point density is extremely high. In case of a typical two-side roof in Figure 2a, an intersecting line by two planes is accurately determined, but the end point of the intersecting line is hard to determine.

An intersection from two crossing lines or an intersecting point from three planes are more desirable types of geometry to determine a conjugate point in Figure 2b. In the case of line crossing, as long as the number of points on the line are reasonable, the mathematical line defined from the multiple points is stable with low uncertainty. Thus, the intersection point from two mathematical lines is a low uncertainty conjugate point. Likewise, if the number of points used for plane modeling is reasonable, the mathematical plane is a stable geometrical feature with low uncertainty. Thus, the three-plane intersection point is a good conjugate point for 3D lidar accuracy assessment. The main reason for the enhanced stability (low uncertainty) is that the geometrical features (line or plane) are constrained by a large number of points, thus it does not suffer from the large uncertainty by a direct pinpointing (corner point, end point, etc.) of the conjugate point.

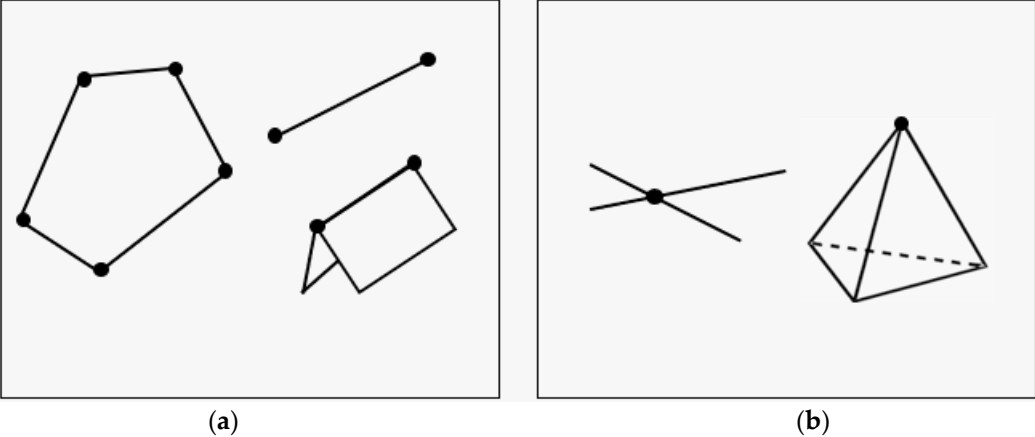

**Figure 2.** Geometrical features: (**a**) Conjugate points with higher uncertainty and (**b**) conjugate points with lower uncertainty.

## 3. Methods

The added uncertainty associated with the conjugate point identification is a critical component in 3D absolute accuracy assessment of the lidar point cloud. We define this additional uncertainty as external uncertainty, which co-exists with inherent positional uncertainty. In this section, we derive a general external uncertainty model for the three-plane intersection point.

### 3.1. External Uncertainty

In the case of typical vertical absolute accuracy (RMSEz) assessments, the measured difference variable $\Delta z$ and its propagated uncertainty $\sigma_z$ in the vertical component are defined as:

$$\Delta z = z - z_0 \tag{1}$$

$$\sigma_z = \sqrt{\sigma_I^2 + \sigma_G^2} \tag{2}$$

where $\sigma_I$ is inherent uncertainty of the lidar data associated with the airborne lidar elevation z, and $\sigma_G$ is ground truth survey uncertainty associated with the true elevation $z_0$. The propagated uncertainty $\sigma_z$ is derived from the following error propagation equation, whose partial derivatives are unity:

$$\sigma_z^2 = \left(\frac{\partial \Delta z}{\partial z} \cdot \sigma_I\right)^2 + \left(\frac{\partial \Delta z}{\partial z_0} \cdot \sigma_G\right)^2 \tag{3}$$

Meanwhile, for the geometrical feature-based 3D absolute accuracy assessment, both ground truth and airborne lidar points are 3D vectors (x, y, z). Thus, the propagated uncertainty will be expressed in full 3D ($\sigma_x$, $\sigma_y$, $\sigma_z$ ). More importantly, airborne lidar conjugate points are characterized by external uncertainty in addition to inherent uncertainty. For each of the three axes, the propagated uncertainty $\sigma$ is expressed as:

$$\sigma = \sqrt{\sigma_E^2 + \sigma_I^2 + \sigma_G^2}, \tag{4}$$

where $\sigma_E$ is an external uncertainty term that is introduced in the full 3D assessment. The reason the external uncertainty term is not included in Equation (2) is that the (x, y) coordinates of the conjugate airborne lidar point were assumed to be the same as those of the ground truth (x0, y0), which considers zero external uncertainty in z, which assumes no uncertainty associated in conjugate point identification. When we perform full 3D accuracy, we must identify conjugate airborne lidar points. The conjugate point identification uncertainty, $\sigma_E$, could be so high that it overpowers the inherent uncertainty of the airborne data. Thus, although we implement the correct strategy of finding conjugate points of the type represented in Figure 2b which gives relatively smaller $\sigma_E$ , it is more

important to quantify how $\sigma_E$ is affected by various conditions. Thus, the investigation of the influence from various conditions is the essence of this paper. The combination of several dominant variables creates unrealistically large parameters to implement when using real airborne lidar data. Thus, this research can be best performed using the airborne lidar simulator [28].

Once the quantitative external uncertainty model is developed, we will face another important question: What should be the maximum allowed external uncertainty? In Equation (2), the uncertainty of the ground truth data $\sigma_G$ must be at least three times better than the uncertainty $\sigma_I$ of the data being tested, which means that $\sigma_z$ in Equation (2) can be approximated as $\sigma_I$ because inside the square root, $\sigma_I{}^2$ is substantially larger than $(\sigma_I/N)^2$ , where N is 3.0 or greater. Assuming the ground truth uncertainty is three times better than that of the inherent uncertainty of airborne lidar, the effect of external uncertainty to the propagated uncertainty in 3D absolute accuracy assessment is illustrated in Figure 3. Using Figure 3 and the external uncertainty model explained in the next section, the maximum allowed external uncertainty will be determined.

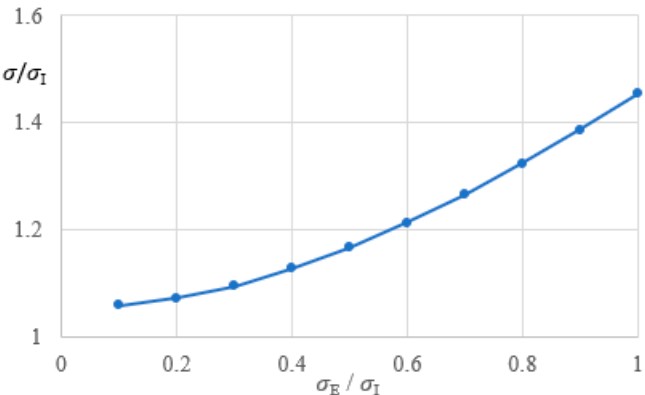

**Figure 3.** Propagated uncertainty as a function of fractional external uncertainty.

## 3.2. Airborne Lidar Simulator

An airborne lidar point cloud simulator [28] was used for the external uncertainty study. The two main components of the simulator are (1) the radiative transfer solution of the lidar waveform and (2) the direct georeferencing of the waveform peak.

A lidar waveform is computed using the laser properties (pulse width, beam divergence angle, and the pulse distribution function), pulse distance determined by sensor altitude and scanner, environmental parameters (absorption and scattering coefficient of the atmosphere), and the 3D geometrical definition of a target [28]. A full waveform solution uses the radiative transfer theory of a laser pulse [29,30]. The radiative transfer theory is numerically solved for the laser beam irradiance distribution function at a given propagation distance. Two irradiance distribution functions are computed: irradiance due to the laser beam propagation and irradiance due to the receiver sensitivity propagation. The irradiance distribution function interacts with a target whose geometry is defined in the 3D coordinate system, and the waveform intensity at a given time is obtained by numerically solving a laser interaction governing equation [29,30].

The second major component of an airborne lidar simulator is the solution of a direct georeferencing equation. A scanner module (scanner type, scan frequency, and the field of view), a global navigation satellite system and strap-down inertial navigation system (GNSS/SINS) module for sensor position and orientation, the flight parameters (sensor altitude, flight speed), and the calibration parameters (boresighting and lever-arm) are the essential inputs required to solve a lidar direct georeferencing equation to estimate the 3D position of a laser-target interaction spot.

To model external uncertainty, we designed specific targets for the 3D uncertainty simulation. A tetrahedron or a pyramid target was created, and a large array of these pyramid targets were simulated as a surface DEM (digital elevation model). As the simulator produces lidar point clouds with various

realistic input parameters, the distribution of the point clouds shows spatially inhomogeneous patterns strongly influenced by the roll, pitch, and heading, as well as the scanner type. All these were necessary to simulate realistic situations for the general external uncertainty model development, as the real targets will be placed at a random position, affecting external uncertainty.

### 3.3. Uncertainty Factors for the Three-Plane Intersection Point

When a conjugate point is determined based on three-plane geometric features, the associated conjugate point uncertainty is derived from three major sources: (1) lidar system precision, (2) point density, and (3) dimension of the geometric feature. These three factors will be used to build an experimental design of the lidar simulation to investigate their influence on the external uncertainty and to create a general external uncertainty model for the three-plane intersection point.

The precision of the lidar system is represented by total propagated uncertainty (TPU). TPU is the propagated statistical uncertainty resulting from all potential uncertainty sources, which are GNSS position, inertial measurement unit (IMU) orientation, boresighting angles, lever arm, scanner parameters, and the lidar ranging uncertainty [28]. Although the TPU is estimated for each axis ($\sigma_x$, $\sigma_y$, $\sigma_z$), it is also practically useful to express TPU by the vertical TPU ($\sigma_z$) and horizontal TPU ($\sigma_h = \sqrt{\sigma_x^2 + \sigma_y^2}$). Also, it is possible to combine the vertical and horizontal TPU to a single value ($\sigma = \sqrt{\sigma_x^2 + \sigma_y^2 + \sigma_z^2}$). In this study, we adapt the concept of smooth surface precision (SSP) to represent the TPU.

To define the SSP, we select a lidar point cloud subset from a locally smooth surface, model a mean plane, and compute the normal distances of all points, SSP is then the standard deviation of the normal distances from the mean plane, as illustrated in Figure 4. This is a vector version of the intra-swath relative vertical accuracy defined in the USGS lidar base specification [27].

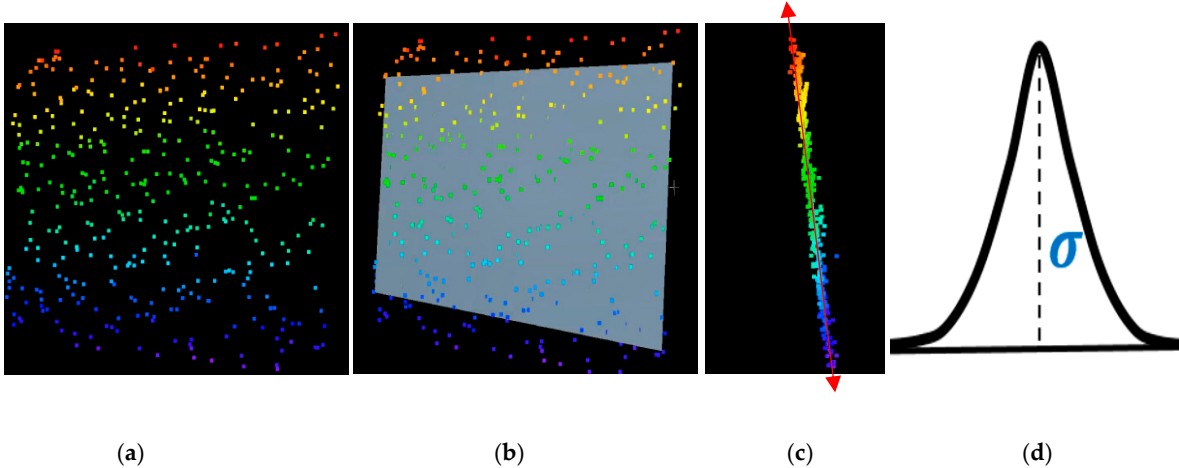

|        (a)        |        (b)        |        (c)        |        (d)        |

**Figure 4.** Concept of smooth surface precision (SSP): (**a**) lidar point cloud, (**b**) plane math model from points, (**c**) side view of the plane that reveals the surface normal distance of lidar points to the plane that is indicated as a red line, (**d**) distribution of surface normal distances and the standard deviation, which is defined as SSP.

When a surface is horizontally flat, the SSP is identical to the TPUz (TPU in $z$-axis, which is the same as $\sigma_z$), and it reflects an increasing horizontal component as the surface is more slanted. In typical airborne lidar data, the off-nadir scanning angle is usually less than $20^\circ$. When a flat surface with arbitrary slant angle is selected, the computed SSP from the flat surface point cloud is a good representative of the TPUz. Thus, in this paper, SSP is used to characterize overall lidar system precision and is one of the three factors used in the experimental design. SSP represents overall lidar system quality in terms of measurement precision.

To model the effect of the three major factors (SSP, point density, and plane area) to the external uncertainty associated with the conjugate point identification, a systematic dataset was generated using an airborne lidar waveform simulator instead of using a real airborne lidar point cloud. The 3D experimental design that corresponds to the three major factors is too large to implement in a practical sense. Instead, we created an experimental plot, building a large array of precisely defined objects of various sizes, deploying many different airborne lidar systems to mimic varying SSP, and tuning system parameters and flight parameters to achieve the varying point density. Since a lidar system usually comes with its optimal setting giving few opportunities to tweak variables, at most allowing a change of scan angle range or laser pulse repetition frequency, the cost and effort would be too high if we did this modeling with real data.

We illustrate only the simulated target object array here because the technical details of the lidar system and the flight parameters that affect the point density are explained in [28]. A large array of 600 pyramid targets were placed between simulated two-swath overlapping areas, representing 3 three-plane intersection points (Figure 5). Pyramids were used instead of tetrahedrons because the most commonly found man-made three-plane objects are immediate derivatives of the pyramid, such as a roof. Also, the single pyramid gives four repetitions of three-plane combinations, which allows the evaluation of 2400 intersection points.

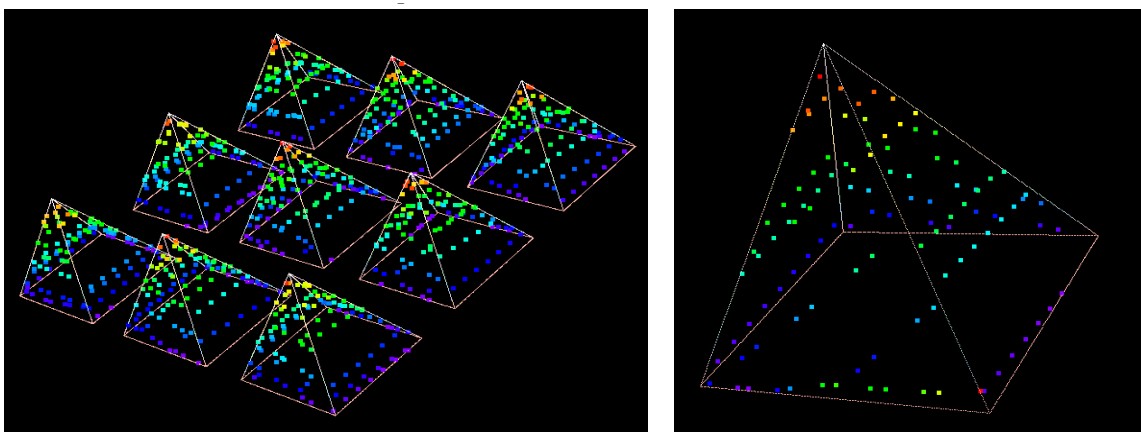

**Figure 5.** Simulated pyramid target array for the study of external uncertainty (showing only 3 × 3 array for illustration instead of the full array of 600 pyramids to visualize the points in detail).

## 4. Results

### 4.1. Simulations of Major External Uncertainty Sources

The first factor, SSP, represents the quality of the airborne lidar system in general. For a given point density and plane area, the higher precision system (low SSP value) will produce point cloud data close to the actual plane, thus plane modeling will be more accurate. Accordingly, the intersection from three planes will be more accurate to the ground truth, while a low precision system produces a noisy point cloud and larger error in estimating the intersection point (Figure 6). Figures 6–8 illustrate horizontal differences ($\Delta x$, $\Delta y$) between an estimated three-plane intersection point and the true point. The blue dots are the distribution of errors ($\Delta x$, $\Delta y$), and $\sigma_h$ is the horizontal standard deviation (also visualized as a red circle) computed from the error distribution.

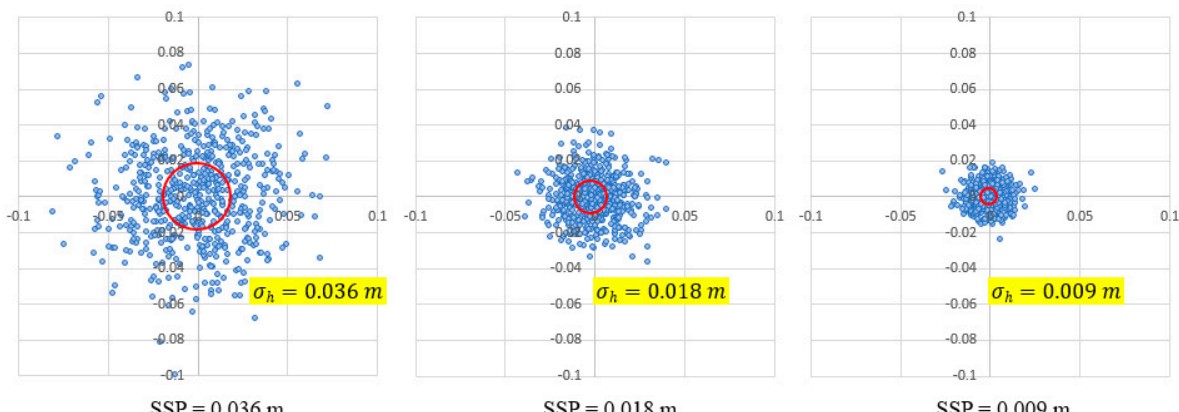

**Figure 6.** Effect of varying SSP to the horizontal external uncertainty $\sigma_h$ for fixed plane size (6 m$^2$) and point density (2 PPSM), where both *x*-axis and *y*-axis are in meter unit.

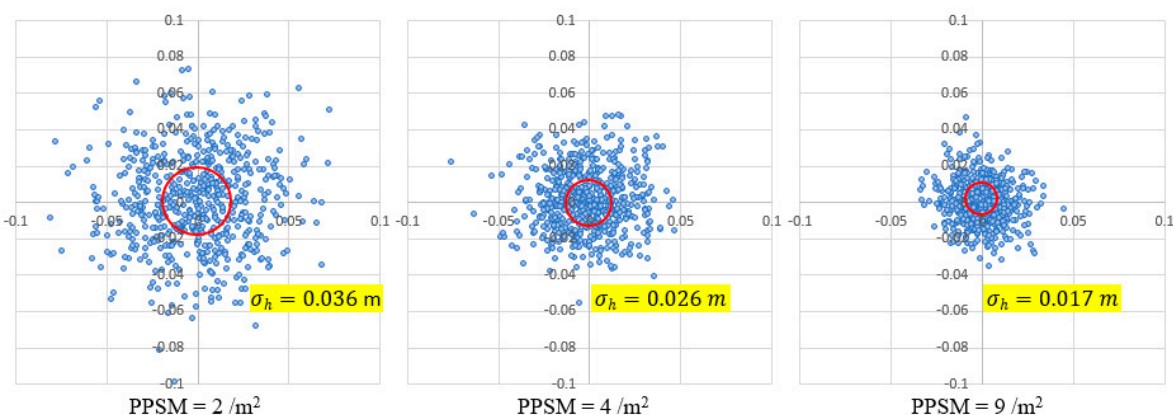

**Figure 7.** Effect of varying point density to horizontal external uncertainty $\sigma_h$ for fixed SSP (0.036 m) and plane area (6 m$^2$), where both *x*-axis and *y*-axis are in meter unit.

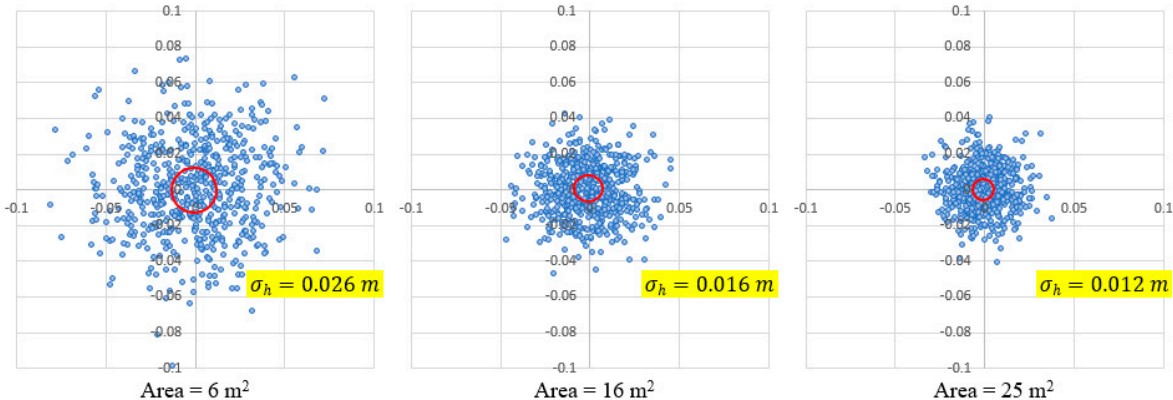

**Figure 8.** Effect of varying plane size (area) to horizontal external uncertainty $\sigma_h$ for fixed SSP (0.036 m) and point density (2 PPSM), where both *x*-axis and *y*-axis are in meter unit.

The second factor, point density, is important because for any given size of the plane, the number of points will be proportional to the point density. The larger number of points in modeling the mathematical plane increases the stability, thus reducing the uncertainty of the plane modeling and eventually reducing the uncertainty of the three-plane intersection point, as shown in Figure 7. Thus, the point density should be one dimension of the experimental design for the external uncertainty modeling.

In a similar manner, the third factor, the area of the plane, is also an important independent variable. For a given point density, the larger the plane, the more points that are available, thus reducing the uncertainty of the three-plane intersection point (Figure 8).

### 4.2. General External Uncertainty Model for the Three-Plane Intersection Point

Only a few small sample results from the large-scale simulation were demonstrated in Figures 6–8. The $\sigma_h$ values in Figures 6–8 can be used to evaluate the external uncertainty and can be provided as a 3D look up table (LUT). However, the LUT is cumbersome to use. To promote the usage of the general external uncertainty model, we suggest the following approach. First, with a fixed SSP value, a two-dimensional (2D) simulation is performed by varying point density and plane area. Each plot in Figure 9 shows the result of the 2D simulation for varying plane areas for each point density, for a fixed SSP. The horizontal axis is expressed as the number of lidar points within a plane, which is computed as a multiplication of point density and plane area. Each dot in the plot represents a 2D simulation combination (point density and plane area). The illustrated cases in Figure 9 are for 2, 4, 9, 16, and 25 PPSM. Each plot occupies a different range of x-axes, because low point density (2 PPSM) needs to simulate for large plane (up to 30 m$^2$), while high point density (25 PPSM) needs to simulate for relatively small planes (4–8 m$^2$). When all plots are combined (the last plot in Figure 9), the results are lined up along a characteristic line with small variability.

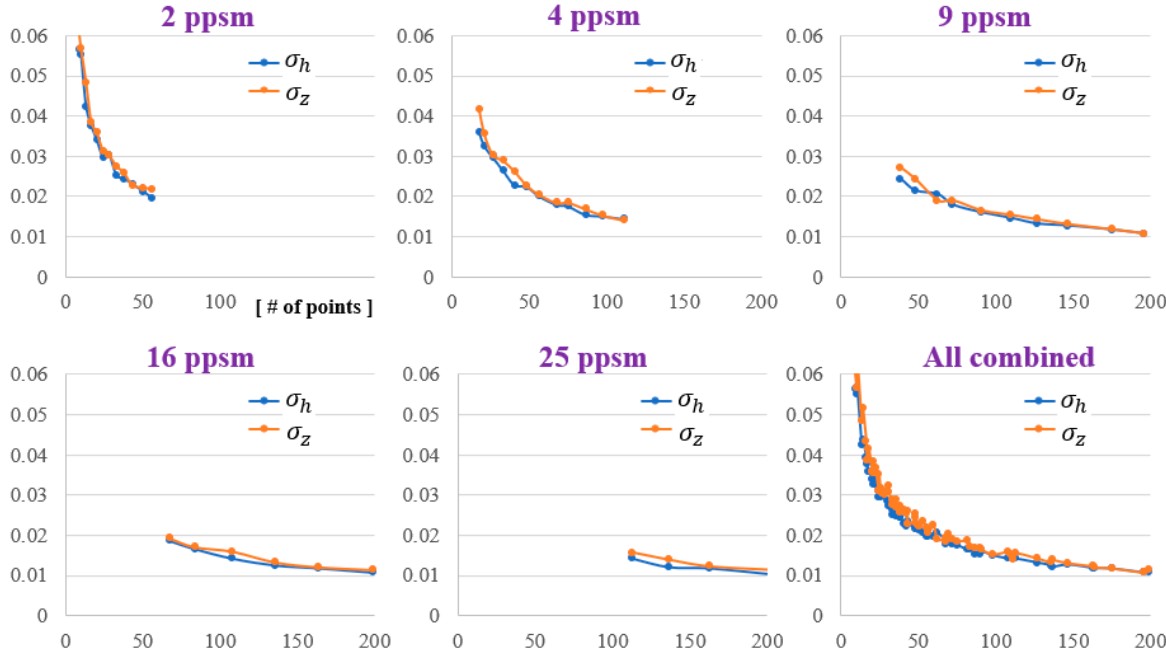

**Figure 9.** Horizontal and vertical external uncertainty $\sigma_h$, $\sigma_z$ in meter unit from two-dimensional (2D) simulation (varying point density and area) with fixed SSP (0.036 m).

The external uncertainty function in Figure 9 is a specific case with SSP of 0.036 m. When many systematic simulations are performed for varying SSPs, we can combine the uncertainty function results in a single plot as shown in Figure 10a. Since the horizontal axis represents the combination of 2D (point density and plane area) and each function (curve) represents each SSP, Figure 10a is the entire 3D LUT of the external uncertainty for a three-plane intersection.

Although the data represented in Figure 10a are a virtually complete 3D LUT and uncertainty for any arbitrary three-factors can be interpolated, it is possible to proceed with one more stage of abstraction of the model. When each curve in Figure 10a is divided by the corresponding SSP value, all four curves are collapsed into a single general curve, as demonstrated in Figure 10b. The vertical axis in Figure 10b represents the "normalized" external uncertainty as a multiple of the SSP. The

actual uncertainty in physical length unit can be achieved by multiplying the specific SSP value. For instance, consider the case of normalized uncertainty 1.0 that is the blue dot in Figure 10b, the actual uncertainty in geometrical length is 7.4 cm for SSP = 0.074 m and 1.8 cm for SSP = 0.018 m, and so on. The dotted curve in Figure 10b is the general external uncertainty model for a three-plane intersection point, which is modeled as a polynomial $(c_0 + c_1 x + \ldots + c_8 x^8)$, where x is the number of the points for a plane modeling and the coefficients $(c_0, c_1, \ldots, c_8)$ are (8.78878, –2.00378, 0.234578, –1.55955×10$^{-2}$, 6.27597×10$^{-4}$, –1.55616×10$^{-5}$, 2.32200×10$^7$, –1.91055×10$^{-9}$, 6.65621×10$^{-12}$).

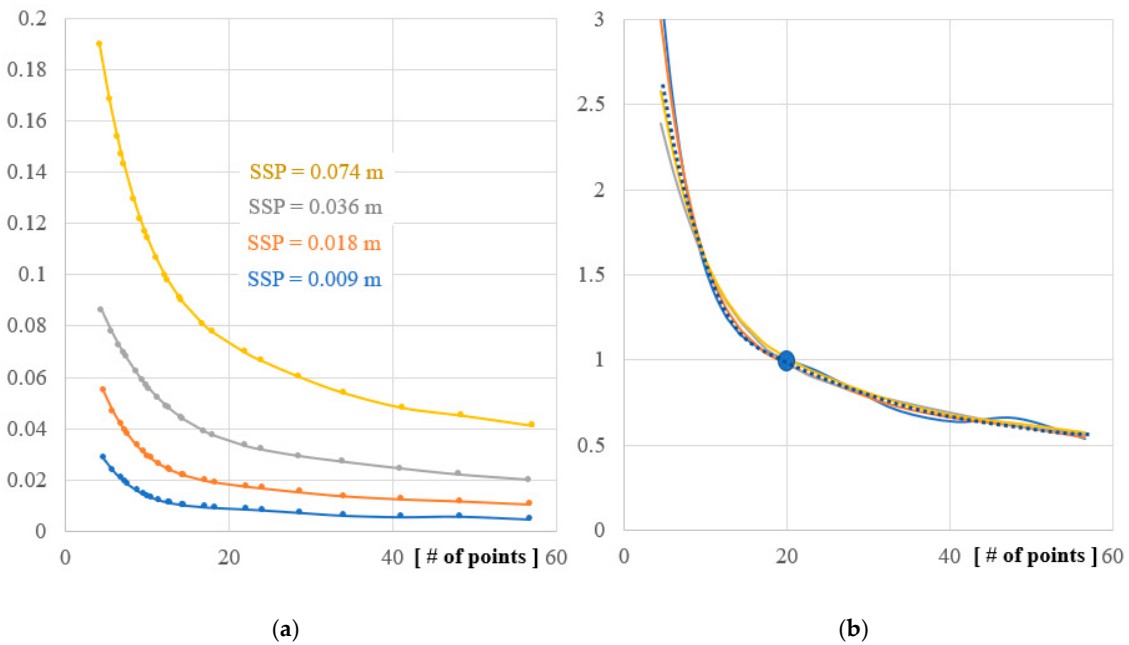

(**a**)    (**b**)

**Figure 10.** (**a**) External uncertainty function in meter unit for varying SSP values, (**b**) General external uncertainty function for three-plane intersection, where the *y*-axis represents the multiple of SSP.

## 5. Discussion

The general external uncertainty function (dotted curve in Figure 10b) can be used to give a specific guideline in geometric feature-based 3D absolute accuracy assessment. The essence of the general external uncertainty model is that it determines whether a specific geometric feature-based conjugate point identification is valid or not. Only the valid conjugate points can be used to compute RMSE in the absolute 3D accuracy assessment.

The Figure 11a is the general external uncertainty function. In the first step, the general external uncertainty model in the SSP unit in Figure 11a needs to be re-scaled to units using specific SSP (e.g., 3 cm), as shown in Figure 11b. Then, the required minimum number of lidar points of a plane to meet the maximum allowed external uncertainty (e.g., 2 cm) gives the minimum number of required lidar points to be used for plane modeling (e.g., 40), as illustrated in Figure 11b. The minimum required number of lidar points needs to be converted to the minimum required area (MRA) of a plane using PPSM of the lidar data. For instance, 20 m$^2$ minimum plane area is needed for 2 PPSM, as shown in Figure 11c. Thus, from our analysis, any conjugate point computed from planes smaller than 20 m$^2$ is invalid.

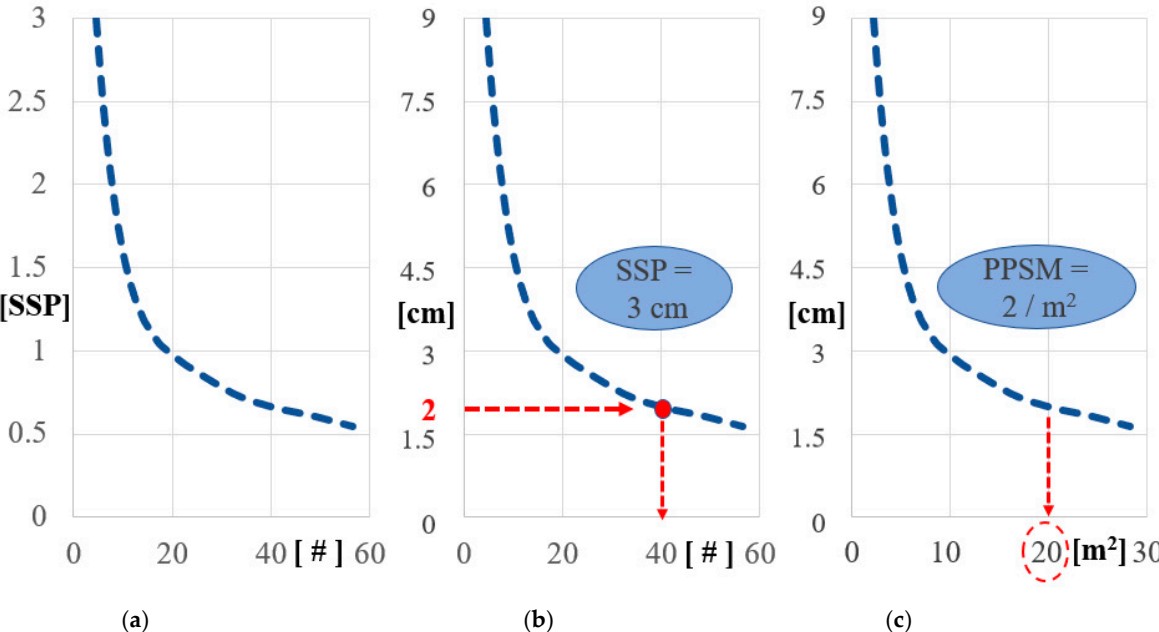

(a)                              (b)                              (c)

**Figure 11.** (**a**) General external uncertainty model, (**b**) re-scaled uncertainty and the maximum allowed uncertainty, (**c**) MRA of a plane.

The maximum allowed external uncertainty can be set based on the general propagated uncertainty curve (Figure 3). This number is usually set by a government agency or lidar authority. For instance, the USGS 3D Elevation Program (3DEP) specifies 10 cm or less of non-vegetated absolute vertical accuracy for the quality level 1 (QL1) data. Based on 10 cm maximum RMSEz, considering the additional challenge for 3D absolute accuracy, if the USGS decides to loosen the requirement by 10%, which is the case of 1.1 in Figure 3, then the propagated uncertainty 1.1 requires the maximum external uncertainty as 0.3 times the maximum inherent uncertainty (10 cm), which is 3 cm. A maximum external uncertainty of 2 cm was used in the example in Figure 11b. Figure 3 gives a basis for any lidar authority to use in making maximum uncertainty requirement decision.

As an example of using a three-plane object for the general external uncertainty model, Figure 12 shows a house with a four-sided roof and a 3 m x 3 m central area used for plane modeling. Figure 12a is the ground truth terrestrial lidar scanner (TLS) data and the image of the house is given in Figure 12b. Three planes were modeled using the TLS points in the gray areas and the two intersecting lines, and a three-plane intersection point is shown in Figure 12c. Three planes using airborne point cloud (large dots in Figure 12d) are also modeled. For easier comparison, the TLS point cloud is still shown as small dots.

As shown in Figure 12b, TLS data have large PPSM and small SSP. The airborne data also have quite high (~20) PPSM, but the relatively high (~3.5 cm) SSP indicates a relatively noisy lidar system. The main question is whether the three-plane intersection point from noisy airborne data is a valid conjugate point or not. The general external uncertainty model is used to answer to this question. We assume maximum allowed external uncertainty tolerance as 3.0 cm. Using 3.5 cm SSP and 3.0 cm tolerance, the general external uncertainty model requires a minimum of 20 points for the valid plane modeling. Since the airborne data are about 20 PPSM, the MRA in plane modeling is about 1 m². All three plane areas are a minimum of 2 m², thus the conjugate point from the three-plane modeling is valid and this point can be used for absolute 3D accuracy assessment. It means the lidar point (x1, y1, z1) in Figure 12e derived from three-plane modeling is a valid conjugate point to the ground truth point (x0, y0, z0) in Figure 12c. Thus, the difference between the two positions can be used as one of the many points to compute the 3D absolute accuracy (RMSEx, RMSEy, RMSEz).

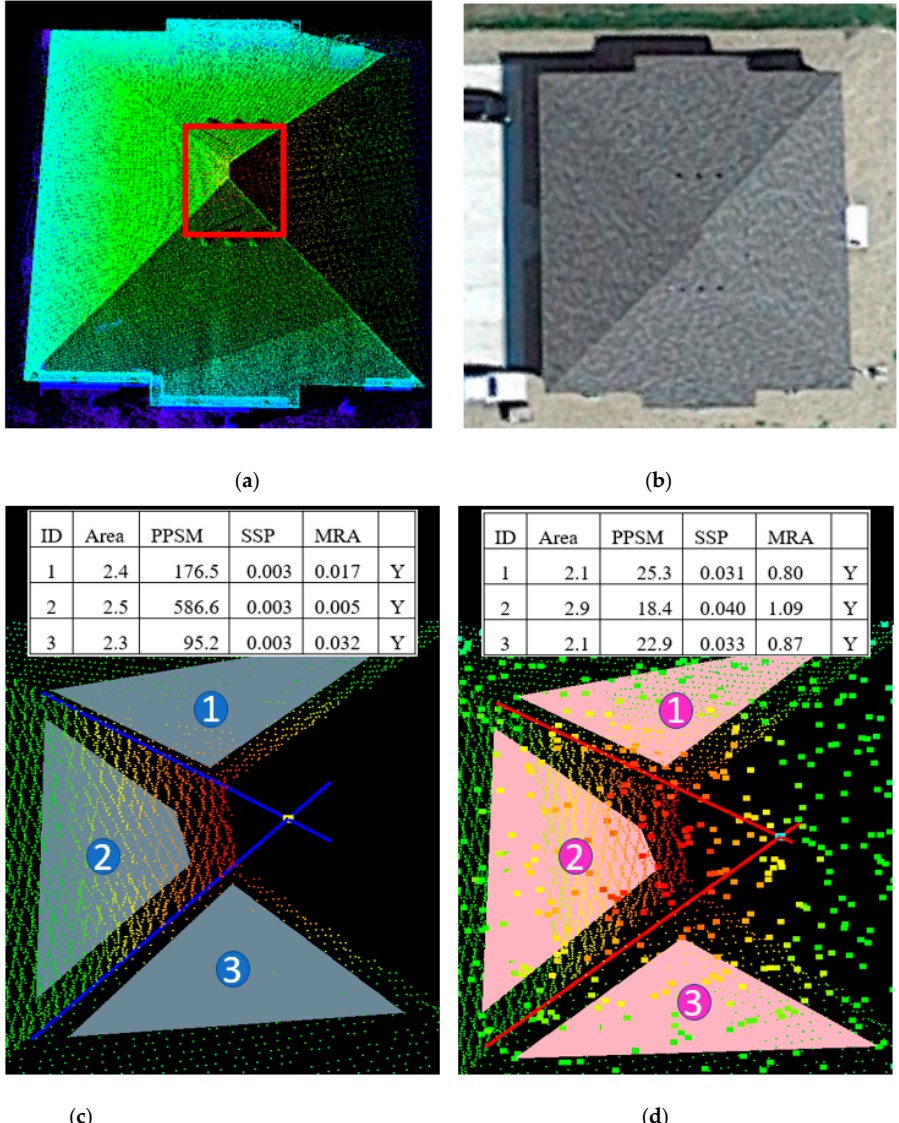

**Figure 12.** (**a**) Ground truth terrestrial lidar scanner (TLS) data for a 4-sided roof object, (**b**) image of the roof, (**c**) TLS data and three-plane modeling, (**d**) airborne point cloud three-plane modeling.

Ground truth is not obtained by a direct measurement in the ground survey, such as measuring the intersection point (e.g., the point in Figure 12c) using total station or another instrument, but one floating in 3D space. The intersection point does not physically exist (Figure 12a,c) or is invisible to the optical instrument (hidden under the ridge, see Figure 13a,b). Thus, the ground truth should be obtained via high point density and a high precision instrument, such as TLS. This is an added complexity compared to the conventional point measurement, such as using GNSS rover or total station. However, it is necessary in pursuing advanced 3D accuracy assessment.

Figure 13 shows an example similar to Figure 12, using another three-plane building object. The quality of the airborne data in terms of SSP is quite low (about 4.4 cm) despite relatively high point density (about 23 PPSM). The general external uncertainty model using 4 cm SSP and the 3.0 cm tolerance gives about 40 as the minimum number of points for the plane modeling. Dividing 40 by the 23 PPSM gives roughly 2 m$^2$ as an MRA for the mathematical plane modeling. Since the area used for the plane modeling is about 4 m$^2$ or larger, the conjugate point determined by this specific plane modeling is valid.

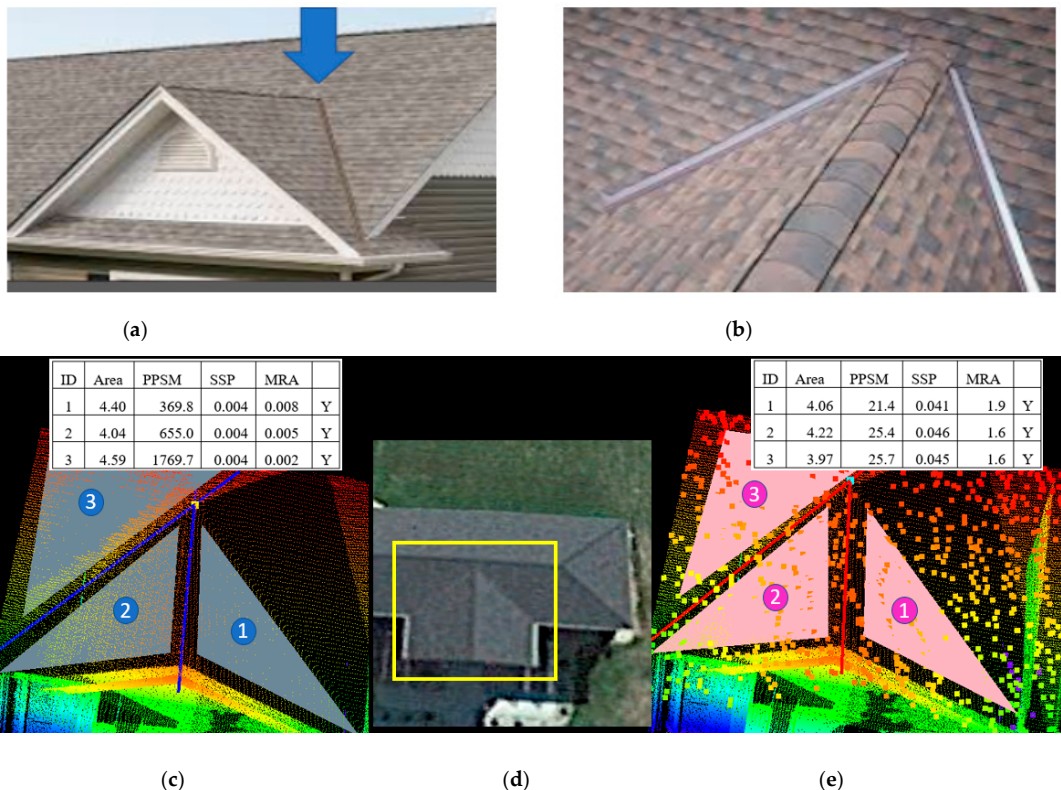

**Figure 13.** (**a**) Three-plane building, (**b**) zoom-in of the three-plane area, (**c**) TLS data and three-plane modeling, (**d**) real building image with the yellow box area used for modeling, (**e**) airborne point cloud three-plane modeling with TLS data.

This specific airborne data are low precision data (about 4.4 cm SSP), and the definition of the plane boundary is poor. Figure 13e shows that many airborne lidar points extended beyond the physical boundary of the roof plane. The poor boundary definition is another separate component needed in the specification of lidar quality. The many examples in Figure 14 demonstrate whether the three-plane intersection point is valid for 3D absolute accuracy analysis by investigating the qualification of each plane using the general external uncertainty model.

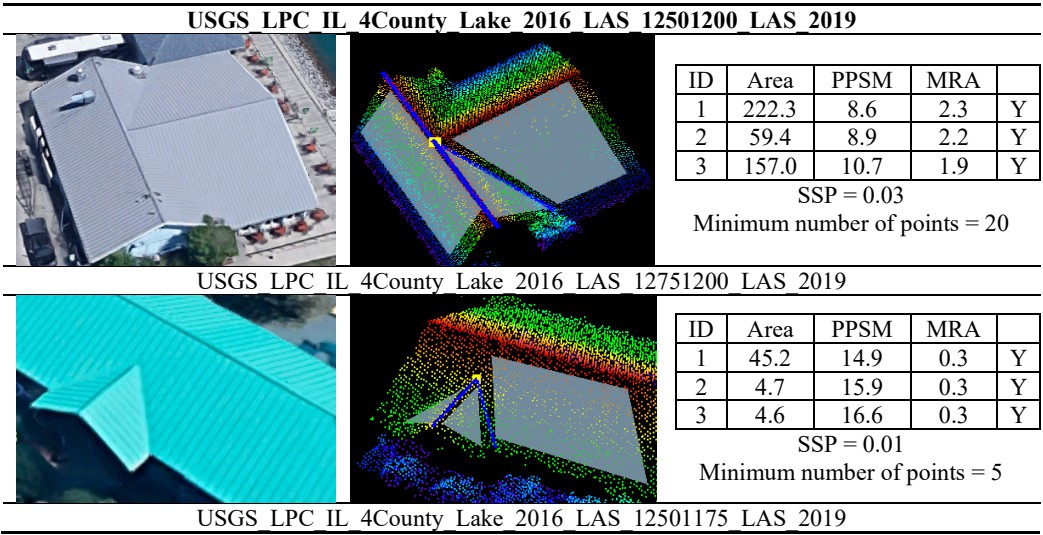

**Figure 14.** *Cont.*

| USGS_LPC_IL_4County_Lake_2016_LAS_12501175_LAS_2019 | | |
|---|---|---|

| ID | Area | PPSM | MRA | |
|---|---|---|---|---|
| 1 | 40.5 | 23.7 | 2.1 | Y |
| 2 | 8.7 | 19.3 | 2.6 | Y |
| 3 | 8.2 | 21.3 | 2.4 | Y |

SSP = 0.05
Minimum number of points = 50

| USGS_LPC_UT_WasatchFault_L3_2013_12TVK4300053000_LAS_2016 | | |
|---|---|---|

| ID | Area | PPSM | MRA | |
|---|---|---|---|---|
| 1 | 35.7 | 13.1 | 1.5 | Y |
| 2 | 73.8 | 12.8 | 1.6 | Y |
| 3 | 48.9 | 12.0 | 1.7 | Y |

SSP = 0.03
Minimum number of points = 20

| USGS_LPC_UT_WasatchFault_L3_2013_12TVK4300053000_LAS_2016 | | |
|---|---|---|

| ID | Area | PPSM | MRA | |
|---|---|---|---|---|
| 1 | 37.3 | 7.2 | 2.8 | Y |
| 2 | 85.2 | 7.4 | 2.7 | Y |
| 3 | 99.1 | 7.2 | 2.8 | Y |

SSP = 0.03
Minimum number of points = 20

| VA-WV-MD_FEMA_Region3_UTM18_2012_001186 | | |
|---|---|---|

| ID | Area | PPSM | MRA | |
|---|---|---|---|---|
| 1 | 26.7 | 2.2 | 22.7 | Y |
| 2 | 28.8 | 2.1 | 24.0 | Y |
| 3 | 19.3 | 1.9 | 10.4 | Y |

SSP = 0.05
Minimum number of points = 50

| VA-WV-MD_FEMA_Region3_UTM18_2012_001186 | | |
|---|---|---|

| ID | Area | PPSM | MRA | |
|---|---|---|---|---|
| 1 | 26.2 | 2.1 | 23.2 | Y |
| 2 | 14.3 | 2.2 | 22.9 | N |
| 3 | 12.9 | 2.0 | 24.5 | N |

SSP = 0.05
Minimum number of points = 50

| SD_MinnehahaCo_2008_000076 | | |
|---|---|---|

| ID | Area | PPSM | MRA | |
|---|---|---|---|---|
| 1 | 124.4 | 0.6 | 95.0 | Y |
| 2 | 134.2 | 0.4 | 127.3 | Y |
| 3 | 167.5 | 0.6 | 96.0 | Y |

SSP = 0.06
Minimum number of points = 55

| SD_MinnehahaCo_2008_000077 | | |
|---|---|---|

**Figure 14.** *Cont.*

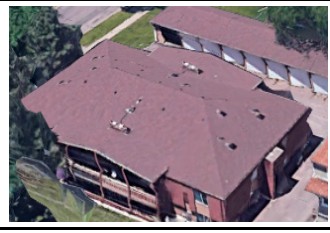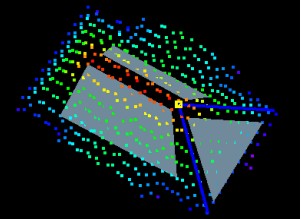

| ID | Area | PPSM | MRA | |
|----|------|------|------|---|
| 1 | 57.1 | 0.6 | 84.9 | N |
| 2 | 129.5 | 0.6 | 97.5 | Y |
| 3 | 53.5 | 0.5 | 105.2 | N |
| SSP = 0.06 | | | | |
| Minimum number of points = 55 | | | | |

**Figure 14.** Example usages of general external model. The external uncertainty tolerance of 3 cm was assumed in these examples. The three-dimensional Elevation Program (3DEP) source point cloud product identifier for each example can be used to obtain the data from the U.S. Geological Survey (USGS), The National Map (TNM).

In Figure 14, example data from 3DEP source lidar point cloud available from the USGS, The National Map (TNM), are illustrated. The examples include various system precision, point density, and plane size. A specific combination shows whether the plane is valid for intersection point computation.

## 6. Conclusions

As the point density of the airborne lidar increases, the need for the full 3D absolute accuracy assessment of the lidar point cloud is gaining more attention. We described the difficulty of identifying a conjugate point of the ground checkpoint represented in the airborne lidar point cloud. A suggested solution is to use geometric feature-based conjugate point identification. However, the uncertainty associated with this type of identification can vary. Thus, the real questions in practice include what are the preferred geometric features and what are the valid conditions for the conjugate point?

This paper documents extensive airborne lidar simulation modeling with a large array of pyramid targets in order to estimate the uncertainty associated with identifying a conjugate point, which we called external uncertainty. We explained a general external uncertainty model for the three-plane intersection point. We also demonstrated the practical use of the general external uncertainty model using several example lidar point cloud data. The development of the external uncertainty model is a crucial component in establishing a foundation for the 3D absolute accuracy assessment of the lidar point cloud.

A full-scale 3D absolute accuracy assessment will find a statistically meaningful number of three-plane intersection targets and matching high-precision survey data of the same targets. Each target object from the airborne lidar point cloud will be tested for qualification using the external model. The differences of intersection points between airborne and survey data will be used for accuracy statistics. In practice, finding only the three-plane intersection points will be somewhat limiting. In the future, other external uncertainty models for other geometric feature-based targets, such as elevated line-crossing or intensity-based line-crossing, will be presented. Thus, the 3D absolute accuracy assessment in practice will utilize several different types of targets.

**Author Contributions:** Conceptualization, M.K.; methodology, M.K.; software, M.K. and S.P.; validation, S.P.; formal analysis, M.K., S.P.; investigation, J.I.; resources, J.D.; data curation, M.K., J.D. and J.I.; writing—original draft preparation, M.K., J.S.; writing—review and editing, J.S. and J.N.; visualization, S.P.; supervision, G.S.; project administration, G.S.; funding acquisition, G.S.

**Funding:** This research was funded by the USGS National Geospatial Program (NGP) 3D Elevation Program (3DEP).

**Conflicts of Interest:** The authors declare no conflict of interest.

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
