# Peer review of "General External Uncertainty Models of Three-Plane Intersection Point for 3D Absolute Accuracy Assessment of Lidar Point Cloud"

_remotesensing, doi:10.3390/rs11232737_

Round 1
Reviewer 1 Report
The authors focused on how to identify a conjugate point of a ground-surveyed checkpoint in the LiDAR point cloud with the smallest possible uncertainty value for assessing accuracy in full 3D. This is really a challenging task because of relatively coarse point-spacing in airborne LiDAR. They investigated what are the preferred geometric features and the valid conditions for identifying the conjugate points: they developed a generalized external uncertainty model for the three-plane intersection point identification process, and further demonstrated the practical use of the general external uncertainty model using both real and simulated data. I think that the construction of the generalized external uncertainty model is very meaningful and practical for LiDAR application. The authors have fully analyzed the factors of the external uncertainty and carefully provided the method to utilize the proposed generalized external uncertainty model. As a result, I suggest to accept the manuscript after minor revision. Some suggestions and questions are as follows:
I. Most of the abstract described the importance of the research and the definition of “external uncertainty”. I think that the importance of the research can be simplified, and more of the results can be added to the abstract.
II. Line 37: “Hodgson and Bresnahan (2004)”, the citation format may be improper.
III. Line 71: “…Habib et al diagnosed the systematic errors inherent in the…”, the reference number?
IV. Line 74-76: “Another method investigated to calibrate data involves fitting data to planes. The method 74 proposed in Skaloud et al (2006) estimates the calibration parameters by conditioning a group of 75 points to lie on a common plane [5].” The citation format?
V. The number of references is only 11, I think it is too less. Some other references may help the authors to describe the related works indirectly. For example, in “Huang R, Zheng S, Hu K. Registration of Aerial Optical Images with LiDAR Data Using the Closest Point Principle and Collinearity Equations. Sensors, 2018, 18(6):1770”, check points were extracted from the LiDAR data by using the intersections of two artificial line segments or three artificial planes for accuracy assessment.
VI. Figure 3, Figure 9, and Figure 10: I suggest to explain the meaning of the X-axis and Y-axis in the Figure. I am difficult to comprehend why the figures can combine together to the last sub-figure.
VII. Figure 6~8: I suggest to explain the meaning of the X-axis and Y-axis, and add the units of the values of the axis.
VIII. As the authors said, the external uncertainty for a three-plane intersection point is main influenced by SSP, point density, and the area of the plane. I am confused for the general external uncertainty model for three-plane intersection point described in Figure 10(b), I seem not to find how the area of the plane influence the external uncertainty from the proposed model.
IX. Except for SSP, point density, and the area of the plane, why the authors do not consider the angles of the planes or the relative angles among the three planes?
Author Response
Thanks for your kind review for our paper.
I.more of the results can be added to the abstract.
==> other reviewer also pointed this. We made change accordingly.
II.L37: “Hodgson and Bresnahan (2004)”, the citation format may be improper.
==>fixed
III.L71: the reference number?
==>number added
IV.L74-6: “citation format?
==>format fixed
V.references
==>Thanks for the recommendation, we added a number of references
VI.Figure 3, Figure 9, and Figure 10: I suggest to explain the meaning of the X-axis and Y-axis in the Figure. I am difficult to comprehend why the figures can combine together to the last sub-figure.
==>x-axis: points on a plane, y-axis: horizontal and vertical uncertainty in meter unit, it is specified in the figure caption. Also, the explanation on why figures can combine is added in the text.
VII.Figure 6~8: I suggest to explain the meaning of the X-axis and Y-axis, and add the units of the values of the axis.
==>both x-axis and y-axis are in meter unit and they are the deviation from true horizontal position of the3-plane intersection point. It is remarked in the figure caption.
VIII.As the authors said, the external uncertainty for a three-plane intersection point is main influenced by SSP, point density, and the area of the plane. I am confused for the general external uncertainty model for three-plane intersection point described in Figure 10(b), I seem not to find how the area of the plane influence the external uncertainty from the proposed model.
==>In the Fig(9), two-dimensions (ppsm and plane area) are already combined (ppsm multiplied by area) to the number of points in a plane. Thus, x-axis of Fig (9-10) are the number of points and it includes the varying plane area.
Except for SSP, point density, and the area of the plane, why the authors do not consider the angles of the planes or the relative angles among the three planes?
==>Excellent valid points. These are important consideration and they need to be addressed in the auxiliary required conditions. As of now we assume planes should have moderate-slope and typical relative angles in the perspective of near-nadir airborne lidar view. As long as we avoid near-parallel planes and extreme angle like a vertical wall, we believe these two factors are not major. For instance, larger positioning error of the vertical wall will be compensated by the fact that the “minimum number of points” requirement will ask much larger area of the vertical wall due to the small projected area. In the future papers, we will further expand the list of external models for other types of geometric feature-based 3D objects, such as parapet-type elevated linear crossing, intensity-based parking lot paint line crossing, the simple two-sided roof structure, and elevated and isolated point target. For each of these candidate geometric features, we will present detailed auxiliary requirements on top of the major factors for the external uncertainty model.
Reviewer 2 Report
The subject of the paper is original and adresses quality assurance issues that are not often described. In this sense, the research is original and should be considered for publication.
However, there are some issues with both result presentation and content that should be adressed.
General remarks:
Bibliography is not extensive and the number of citation is relatively low There are problems with the vocabulary used The applicability of the results should be made clearer at the beginning of the paper
Review of the paper:
"General External Uncertainty Models of 3-Plane Intersectoin Point
for 3D Absolute Accuracy Assessment of LiDAR Point Cloud"
l52: Non-vegetated Vertical Accuray (NVA)
l72: instrument and assessed....but they did not...
l74-79: Cite more recent boresight calibration papers:
[1] "Simultaneous Calibration of ALS sytems
and alignment of multiveiw LiDAR scans of urbans area" (M. Helbel U. Stilla), IEEE 2012
[2] "Automatic Data selection and boresight Adjustment of LiDAR system", R. Keyetieu, N. Seube, Remote Sensing 2019
l85-... : cite other papers that contributed to the test of geometric targets (Hexaogonal targets, spheres) in the framework of ALS and/or mobile LiDAR:
[3] Assessing LiDAR accuracy with Hexagonal Retro-reflexive targets, R. Canavosio-Zuzelski et al..., PR&RS 2013
[4] Geomatric validation of a ground based mobile laser scanning system, D; Barber 2007
[5] A robust solution to high accuracy geolocation: quadruple integration of GPS, IMU, pseudolite and TLS, D. Grejner-Brzezinska et al. 2009
l99 ...point clouds can be accomplished
l132: repetition of "are useful"
l135: ..that extermal point of any polygon are hard to ...
l142: COMMENT: "with low uncertainty": You should consider the estimation method to obtain the geometric feature (plane, line). It may not be robust to outliers for instance... What you say may be understood as "all estimation method provide low uncertainty" which is false (especially this standard least squres !)
l144: COMMENT: "with low uncertainty (Again!): I would say "which uncertainty decreases with the number of observed points".
l171: Define sigma_e using appropriate statistical tools
l186: replace "uncertainty" with SD
l187: Justify that you can "safely" approximate 1/9 by 0.
Fig 3: No units; axis are not defined.
l230: Vocabulary problem: Your article deals with a concept of Metrology. As such, I suggest that you use the Standard wording. Exmaple: Instead of TPU (not defined using international standards), use the Combined Standard Measurement Uncertainty. In general, your article should refer to the concepts and wording of the Int. Vocabulary of Metrology.
l254 ...dataset was generated....
Author Response
We accept all of your grammar suggestions. (Line 52, 72, 99, 132, 135, 254)
We made changes accordingly.
We also appreciate your reference suggestions. (Line 74-79, 85)
We added them in the manuscript.
Line 142, 144 (“low uncertainty”):
As opposed to the case of an isolated lidar point whose conjugate point identification has “high uncertainty”, we make a
general argument that the geometric feature-based point has “low uncertainty” in identifying the conjugate point. After
the general remark in this paragraph, in the later chapter we describe the external factors that affects the uncertainty,
and derive a quantified model for the external uncertainty. The specific concerns (geometric feature estimation method,
or its dependence on the number of points) are considered in the modeling.
Thus, we believe the sentence conveys proper context.
Line 171, Define ( sigma_E ) using statistical tool:
We suggest sigma_E term to address the conjugate point identification uncertainty, which is inevitably introduced during the
3D absolute accuracy analysis. This research considers several relevant factors and derive general external model.
Following your suggestion we derived a polynomial semi-analytical model to fit the external model as closely as possible.
Instead of the plot, the polynomial model will enhance the practical usability of the model (at the end of section 3).
Line 187, Justify "safely" approximate 1/9 by 0 :
We agree that “safely” is not a best word in the context. So, we made change accordingly in the manuscript.
Fig 3 : axis units were defined
Line 230, CSU instead of TPU:
“Guide to the Expression of Uncertainty in Measurement”(GUM) by ISO(1995) clearly defines the Combined Standard
Uncertainty (CSU). However, in the oceanographic charting, bathymetric lidar, and topographic lidar community, the
term TPU has been used for several decades in the identical definition as CSU. It is the combined variance from covariance
matrix and the propagated uncertainty via Jacobian matrix formed from direct the geo-referencing equation of sonar or
lidar system. Currently, the ASPRS/USGS lidar working group, NOAA/USACE bathy lidar and sonar community are working
on the standard TPU products. Because of this reason, we appreciate your point, but we believe we should use TPU.
Remark the applicability of the result at the beginning:
In fact another reviewer also suggested to add the result of external uncertainty model in the abstract. Thus, we made
modification of the abstract accordingly.
We are grateful for your kind review and your appreciation on the essence of our approach
to the full 3D absolute accuracy assessment. This paper is the first to address the concept of external uncertainty model
using 3-plane intersection. In the following papers we will develop external uncertainty of conjugate point using other
geometric features in order to enrich the sources of 3D absolute accuracy assessment.
Reviewer 3 Report
This paper represented their new work on full 3D LiDAR accuracy assessment. Comments are as:
Introduction:
Line 73 missing the reference;(Habib) More reference should be included,
Conjugate point identification
No comments.
General external uncertainty model
Line 212, “global navigation satellite system and inertial measurement unit (GNSS/IMU) module”, IMU is a sensor not a navigation system, strap-down inertial navigation system (SINS) is based on the IMU. GNSS/SINS might be appropriate.
Application of the general external uncertainty model using real data
No comments
Author Response
Thanks for your kind review for our paper.
We believe you appreciated the essence of our approach to the full 3D absolute accuracy assessment.
Line 73 missing reference was fixed and we added a number of references as you suggested.
We appreciate your suggestion of the GNSS/SINS. We made change accordingly.
Reviewer 4 Report
The paper is well written and presented.
As explained in the conclusion, "In practice, finding only the three-plane intersection points will be somewhat limiting. In the future, other external uncertainty models for other geometric feature-based targets, such as elevated line-crossing or intensity-based line-crossing, will be presented". it would have been more interesting seeing comparison and analysis not only on rooftop intersection points but also on intersection points placed closer to the ground, for which it could have been easier to survey the coordinates with different instruments (total station, GNSS).
In this way, the definition of the ground truth could be more accurate. As said by the authors in the conclusions, it is desirable to see further analysis.
I also suggest a review of the references, since most of them are not recent.
Author Response
Thanks for your kind review for our paper.
We believe you appreciated the essence of our approach to the full 3D absolute accuracy assessment.
As you pointed out, we will further expand the list of external models for other types of geometric feature-based 3D objects,
such as parapet-type elevated linear crossing, intensity-based parking lot paint line crossing. As you suggested, the
intensity-based parking lot paint line crossing is a good candidate in the perspective of the ease of ground survey.
We will also perform extensive research to create an external model for the simple two-sided roof structure, which is
commonly found. However, the requirement guideline will be even more complicated than 3-plane intersection, despite
the relative simplicity.
We added a number of references including recent ones.
Round 2
Reviewer 1 Report
The authors have answered all my questions, and I think that the manuscript has been greatly improved. So I suggested to accept the manuscript. However, I also suggest that authors to check the manuscript as carefully as they can before the final publication, for example, the captions of the Figure 12(b) may be still confused.
Author Response
It was a mistake after adding an image carelessly.
The Figure caption Fig 12a, 12b were corrected.
The text at lines (386-388) were also corrected accordingly.
Also, the following minor corrections were made.
A few words were added in Fig 1b caption for better understanding.
Minor centering of Fig 4 ( c ) and ( d ).
Formatting error was corrected for the lines (344 – 355)
Thanks so much for the comment.